# Towards Machine Learning-based Fish Stock Assessment

Stefan Lüdtke
stefan.luedtke@uni-rostock.de
Institute for Visual & Analytic Computing
University of Rostock
Germany

Maria E. Pierce
maria.pierce@thuenen.de
Thünen Institute of Baltic Sea Fisheries, Rostock
Germany

## ABSTRACT

The accurate assessment of fish stocks is crucial for sustainable fisheries management. However, existing statistical stock assessment models can have low forecast performance of relevant stock parameters like recruitment or spawning stock biomass, especially in ecosystems that are changing due to global warming and other anthropogenic stressors.

In this paper, we investigate the use of machine learning models to improve the estimation and forecast of such stock parameters. We propose a hybrid model that combines classical statistical stock assessment models with supervised ML, specifically gradient boosted trees. Our hybrid model leverages the initial estimate provided by the classical model and uses the ML model to make a post-hoc correction to improve accuracy. We experiment with five different stocks and find that the forecast accuracy of recruitment and spawning stock biomass improves considerably in most cases.

## CCS CONCEPTS

• **Applied computing** → **Environmental sciences**; • **Computing methodologies** → *Machine learning*.

## KEYWORDS

fish stock assessment, gradient-boosted trees

**ACM Reference Format:**
Stefan Lüdtke and Maria E. Pierce. 2023. Towards Machine Learning-based Fish Stock Assessment. In *Proceedings of Fragile Earth: AI for Climate Sustainability - from Wildfire Disaster Management to Public Health and Beyond (Fragile Earth '23)*. ACM, New York, NY, USA, 6 pages. https://doi.org/XXXXXXX.XXXXXXX

## 1 INTRODUCTION

Stock assessment is a fundamental component of fisheries management, providing information on the status of fish stocks and allowing for informed decisions on sustainable harvest levels. The gold standard models for stock assessment are age-structured state-space models like SAM [13]. These models explicitly estimate noise in the observations and system dynamics (mortality and recruitment, i.e., the number of individuals leaving and entering the fishable population). These models are then fit to available observations, i.e.

surveys and age-stratified catch data of commercial fisheries. Such models are routinely used for stock assessment of stocks for which sufficient and reliable data is available, like Western Baltic cod.

However, these models need to make strong assumptions about the parametric form of the involved distributions, e.g. by assuming a Beverton-Holt stock-recruitment model or a log-normal observation distribution [1]. The predictive performance of these models can be limited when these assumptions do not hold, or when the behavior of the stock depends on conditions that are not included in the model, like environmental factors or the abundance of other species. Recently, there have been some prominent examples where these models failed to provide reliable assessments. For example, for the Eastern Baltic cod, no quantitative stock assessment has been available since 2014, as existing models could not explain trends in available biological data of this stock [5]. It is assumed that warming and eutrophication have led to qualitative changes in the ecosystem [3], leading to a new dynamic of this stock that is not appropriately captured by existing stock models. With ecosystems changing due to global warming and other anthropogenic stressors, flexible and accurate stock assessment models that are able to incorporate these new influences on the ecosystem are required.

To meet this challenge, machine learning (ML) methods have been explored for stock assessment. In principle, ML models offer greater flexibility in learning non-linear relationships. For example, [4, 9, 10] present models that estimate recruitment based on the spawning stock biomass (SSB), environmental and climate data. Instead of forecasting, all of these approaches focus on quantifying the effect of environmental data on stock parameters. Specifically, they use the SSB times series as model input, which can only be reliably estimated *retrospectively* via classical stock assessment models like SAM. Thus, they cannot be used for forecasts as required for fisheries management.

In this paper, we propose an ML approach that only uses data available in the assessment year, and thus can be used for *forecasting* relevant stock parameters. As training data is scarce, we propose a hybrid model: We fit a statistical stock assessment model (SAM) to the available data, and then make a post-hoc correction of the initial SAM estimate of stock parameters via gradient-boosted trees. This approach is motivated by *hybrid* ML models, which combine mechanistic models that encode domain knowledge (as SAM does in our case) with data-driven ML to model non-linear patterns in the data [6, 18].

We experiment with five different stocks from the Baltic Sea, North Sea and North Atlantic, covering different population dynamics. We show that our approach has a lower forecast error for recruitment and SSB in most cases, compared to standard SAM forecasts. These results show that using ML methods for stock

forecasting is a promising idea and deserves more attention in the future.

## 2 METHODS

We start by briefly reviewing the state-of-the-art statistical models for stock assessment, and afterwards present our hybrid model which combines these models with gradient boosted trees.

### 2.1 State-Space Models for Stock Assessment

State-Space models (SSMs) are hierarchical, statistical models of two time series: First, the *process* time series which reflects the true, but unobserved state. In stock assessment, this is the (age-stratified) abundance of fish individuals. We denote the (true) number of individuals of age $a$ in year $t$ as $n_{a,t}$ and the vector of all age-specific abundances at year $t$ as $n_t$. These variables form a Markov chain, i.e. $n_t$ only depends directly on $n_{t-1}$ via a distribution $p(n_t \mid n_{t-1})$. Variability in this relationship arises due to randomness and uncertainty in mortality and recruitment processes. The second time series are *observations*, e.g. sampling from commercial catches and information from surveys about quantities and age-distributions of fish. We denote the vector of observations at year $t$ as $c_t$. The observations are assumed to depend on the process state via a distribution $p(c_t \mid n_t)$, where variability arises due to imprecision and randomness in the sampled data.

An SSM is fit to an observation time series by estimating the parameters of the process and observation distribution such that they best explain the observed data. We denote an SSM fitted with observation data $c_1, \ldots, c_t$ as $M_t$. This model is then used to estimate the process state, as well as for making forecasts of the process state. We denote the estimate of the process state at time $t_1$ from a model $M_{t_2}$ as $\hat{n}_{t_1}^{t_2}$. When $t_1 = t_2$, then $\hat{n}_{t_1}^{t_2}$ is the *current-year* estimate of the abundance, and when $t_1 = t_2 + k$, then $\hat{n}_{t_1}^{t_2}$ is a *k-year* forecast. From these estimates, stock parameters required for management decisions are derived, like the spawning stock biomass or recruitment.

SSMs in ecology (see [2] for an introduction) use parametric distributions to model the process and observation model. We focus on SAM [13], a specific instance of an SSM for stock assessment. In addition to the age-stratified abundance, SAM explicitly models natural and fishing mortality, recruitment, weights, and catch-at-age data.

### 2.2 Stock Assessment and Forecasting as Supervised Machine Learning

In this section, we show how stock parameter estimation and forecasts can be formulated as supervised learning problems. In principle, these tasks can be seen as time series regression problems, where we need to estimate a stock parameter of interest $r_t$ or $r_{t+1}$ (e.g., the recruitment or SSB) based on a sequence of observations $c_1, \ldots, c_t$. However, this is a challenging problem, as the amount of training data is very limited. Even for well-monitored stocks, the observation time series typically spans < 50 years. Instead, a common approach in such low-data situations is to make use of existing domain knowledge, such that re-learning all (known) dependencies from the data from scratch is avoided [18]. In our case, the modeling assumptions in SAM provide such domain knowledge.

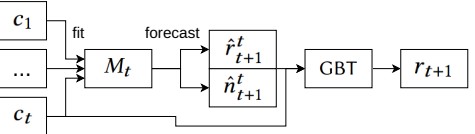

**Figure 1: Model architecture for stock parameter forecasting. To forecast a stock parameter $r_{t+1}$ from observations $c_1, \ldots, c_t$, we first fit a SAM model $M_t$ to the observations. This model is used make an initial forecast of the stock parameter $\hat{r}_{t+1}^t$ and hidden variables (abundances) $\hat{n}_{t+1}^t$. These forecasts and the observation $c_t$ are then used by a gradient boosted tree (GBT) model to compute a corrected forecast $r_{t+1}$.**

Thus, we propose a hybrid approach that utilizes the initial SAM prediction and uses an ML model to correct this initial estimate, shown in Figure 1. More specifically, we propose two variants of our model. For the task of *estimating* the current-year stock parameters, we use SAM estimates of process state $\hat{n}_t^t$, SAM estimates of the stock parameter of interest $\hat{r}_t^t$, as well as the observations $c_t$ as input of an ML model, and the (corrected) stock parameter as output, such that the ML model is representing a function

$$f_1 : \hat{n}_t^t \times \hat{r}_t^t \times c_t \to r_t \tag{1}$$

For the task of *forecasting* the stock parameters, we use a similar setup, but compute SAM forecasts $\hat{n}_{t+1}^t$ and $\hat{r}_{t+1}^t$ as input to the ML model, and the (corrected) forecast $r_{t+1}$ as output, such that the ML model represents the function

$$f_2 : \hat{n}_{t+1}^t \times \hat{r}_{t+1}^t \times c_t \to r_{t+1} \tag{2}$$

Direct measurements of the *true* recruitment or SSB $r_t$, as required for supervised learning, cannot be made. Instead, we use the estimates of a SAM model fitted with data up to the final year $T$ as training data, i.e. $r_t \approx \hat{r}_t^T$. This approach relies on the common assumption in stock assessment that the model estimates become more stable and converge to the true values after a few years of additional observations. The validity of this assumption is visualized in Figure 2. The figure displays the recruitment and SSB estimates generated by all models $M_1, \ldots, M_T$. It can be seen that only the estimates for the *end* of each time series are corrected when additional data becomes available, while the estimates for the previous years remain stable.

### 2.3 Experimental Evaluation

Goal of the experiments was to compare the stock parameter estimation and forecast performance of our hybrid model and the SAM baseline model. We compared our approach to a SAM model as baseline on five stocks, two target stock parameters (recruitment and spawning stock biomass), and two tasks (current-year estimation and forecasting). In the following, the experimental evaluation is described in more detail.

*Stocks.* We evaluated our approach on five different stocks, which are diverse in terms of status (with all stocks currently in decline, but with different severity and confidence), habitat and ecology: Western Baltic cod in ICES subdivisions 22–24 (*Gadus morhua*, WBC); North Sea whiting (*Merlangius merlangus*); plaice in the

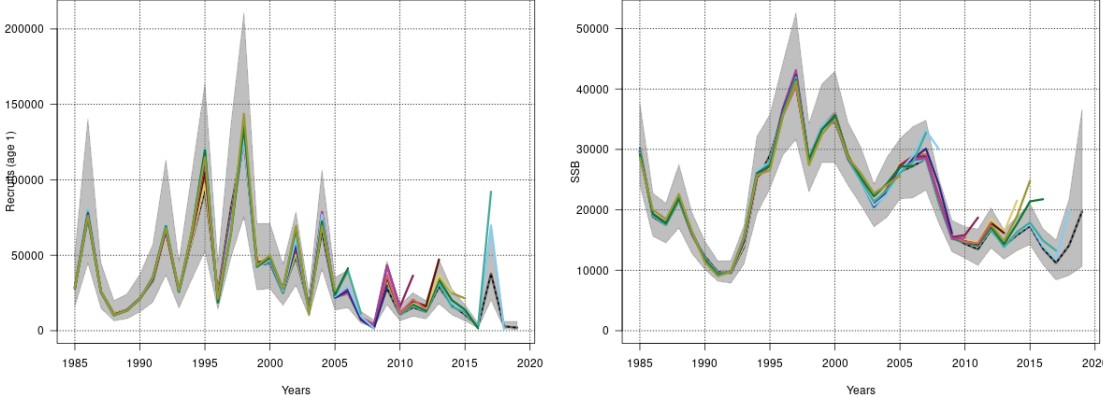

**Figure 2: Retrospective estimates of recruitment and SSB of Western Baltic cod. Each color represents estimates of a different model $M_t$.**

Celtic Sea / Bristol Channel, i.e., ICES divisions 7.f and 7.g (*Pleuronectes platessa*); ling in Faroes waters, i.e. in ICES division 5.b (*Molva molva*); and cod in the Norwegian and Barents Sea north of 67°N (*Gadus morhua*, NCC).

*Target Stock Parameters.* We focused our analysis on two key stock parameters that are crucial for stock management: recruitment and spawning stock biomass. Here, spawning stock biomass is the combined weight of all individuals that have reached sexual maturity and are capable of reproducing, and recruitment represents the number of new young fish entering the population in a given year. Forecasting other stock parameters like fishing mortality (or even natural mortality, which is regularly assumed to be constant in stock models) is subject to future work. We investigated both the current-year estimation (Equation 1) and forecasting (Equation 2) of the stock parameters as tasks.

*Features.* To avoid overfitting or non-convergence of the ML models, we selected different features susbsets for both target variables, which we assumed to be informative for the corresponding target: For the models predicting recruitment, we used the SAM estimate of age-structured abundance $\hat{n}_t$ and the corresponding observations $c_t$ as features. For the models predicting SSB, we experimented with two feature subsets and report the model with lower RMSE: both the SAM estimate of SSB $\hat{r}_t$ and the corresponding observations $c_t$, or only the SAM estimate of SSB $\hat{r}_t$.

*Models and Hyperparameters.* As baseline model, we used the SAM estimates $\hat{r}_t^t$ or forecasts $\hat{r}_{t+1}^t$ (i.e., the first part of the model shown in Figure 1). We used the R implementation of SAM, version 0.12.0 [13]. As machine learning model, we used Gradient-Boosted Trees with the lightGBM package [7] for R. Due to the limited number of training samples, it was not feasible to use a separate validation set for model comparison. Therefore, we did not optimize hyperparameters or experimented with different classifiers. Instead, we chose the hyperparameters upfront according to best practices for such a low-data regime, and used identical hyperparameters for all models. We chose num_leaves = 3, max_depth =

3, min_data_in_leaf = 1, learning_rate = 0.1, and nrounds=60. All other hyperparameters were set to their default value. This way, our evaluation provides a conservative estimate of model performance without overfitting to the test data.

*Evaluation.* To ensure unbiased evaluation of the models, we made sure that (a) the models were only evaluated on data not seen during training, and (b) the models were only trained with data from the past (in contrast to a leave-one-out strategy), to prevent leaking information from the future to the model which would not be available in practice and could overestimate model performance. Therefore, we used the following procedure: for each year $t$, we trained a lightGBM model on data up to time $t-1$ and evaluated its performance on test sample of time $t$. More formally, for each $t$, the training data consisted of tuples $\{((\hat{n}_i^i, \hat{r}_i^i, c_i), r_i)\}_{i=1}^{t-1}$, and the test data was the tuple $((\hat{n}_t^t, \hat{r}_t^t, c_t), r_t)$. Similarly, for the forecast task, for each $t$, we trained a model on data $\{((\hat{n}_{i+1}^i, \hat{r}_{i+1}^i, c_i), r_{i+1})\}_{i=1}^{t-1}$ and evaluated its performance on $((\hat{n}_{t+1}^t, \hat{r}_{t+1}^t, c_t), r_{t+1})$. This evaluation process was repeated for each $t \in \{T-k, \ldots, T\}$, where $T$ is the last year for which data is available and $k$ varies between 17 and 20 years for the different stocks, depending convergence of the corresponding SAM model $M_{T-k}$. All predictions were collected and the root mean squared error (RMSE) as well as the coefficient of determination ($R^2$) were computed as test performance measures.

## 3 RESULTS

*Forecasting stock parameters.* Table 1 (left) shows the RMSE and $R^2$ values for the *recruitment* forecasting task of both the SAM and ML model. The ML model improved the SAM forecast for all five stocks in terms of RMSE, and for three out of five stocks in terms of $R^2$. However, recruitment forecasting is still a challenging task, which can be seen from the overall low $R^2$ values, indicating poor correlation between true and forecasted recruitment. In contrast, forecasting SSB is comparably less challenging. Table 1 (right) shows the evaluation results for this task. The ML model

| | Recruitment forecasting | | | | SSB forecasting | | | |
|---|---|---|---|---|---|---|---|---|
| | ML | | SAM (baseline) | | ML | | SAM (baseline) | |
| Stock | RMSE | $R^2$ | RMSE | $R^2$ | RMSE | $R^2$ | RMSE | $R^2$ |
| WBC | **13970** | **0.383** | 32265 | 0.106 | **5269** | **0.365** | 8218 | 0.009 |
| Whiting | **791179** | **0.101** | 908712 | 0.063 | 63173 | **0.755** | 45258 | 0.666 |
| Plaice | **4665** | 0.165 | 4741 | **0.617** | **768** | **0.599** | 3346 | 0.367 |
| Ling | **848** | **0.165** | 1660 | 0.146 | 3293 | **0.680** | 3772 | 0.554 |
| NCC | **5362** | 0.063 | 15043 | **0.070** | 13287 | **0.278** | 26233 | 0.127 |

Table 1: Results on the *forecasting* task. WBC: Western Baltic cod; NCC: cod in Norwegian Sea and Barents Sea.

| | Recruitment estimation | | | | SSB estimation | | | |
|---|---|---|---|---|---|---|---|---|
| | ML | | SAM (baseline) | | ML | | SAM (baseline) | |
| Stock | RMSE | $R^2$ | RMSE | $R^2$ | RMSE | $R^2$ | RMSE | $R^2$ |
| WBC | **11604** | **0.659** | 16293 | 0.555 | 5962 | **0.749** | **4844** | 0.578 |
| Whiting | 683027 | **0.571** | 575811 | 0.533 | 36012 | 0.812 | **26190** | **0.868** |
| Plaice | 4161 | 0.136 | **3241** | **0.805** | **668** | **0.657** | 2548 | 0.481 |
| Ling | **774** | **0.309** | 1224 | 0.262 | 2620 | **0.784** | **2466** | 0.722 |
| NCC | 5497 | 0.038 | 12145 | **0.213** | **11570** | **0.159** | 34606 | 0.026 |

Table 2: Results for the current-year estimation task. WBC: Western Baltic cod; NCC: cod in Norwegian Sea and Barents Sea.

improves the SAM forecast for all five stocks (with the sole exception of the RMSE value on the whiting stock). The ML models' most significant improvement is observed for Western Baltic cod, where the ML model improves the SAM forecast from no explained variance ($R^2 < 0.01$) to a poor but non-zero variance explanation ($R^2 > 0.35$). Although the improvement for the other stocks is less dramatic, it is still consistently present. These promising results suggest that even our simple approach can significantly improve forecast performance. Note that this improvement is present despite the fact that the evaluation intentionally considered only a single ML model for each stock without hyperparameter optimization to avoid optimizing for test set performance.

*Current-year estimation of stock parameters.* The results for the current-year estimation are shown in Table 2. Generally, current-year estimation is a simpler task than forecasting, as data more directly related to the estimated quantity (i.e., from the same year) is available. This can be seen from the overall smaller RMSE and larger $R^2$ values for all stocks and models, compared to the forecasting task. Here, the ML approach improves recruitment estimation performance on three stocks, and SSB estimation performance on four of the five stocks (measured by $R^2$), overall being consistent with the findings for the forecasting task.

*Feature Importance.* To gain additional insight into the models, we plotted SHAP feature importance values [11] for the recruitment and SSB estimation models for the Western Baltic cod stock (Figure 3). It can be seen that for recruitment estimation, the observations and SAM-estimated abundances of age-1 fish are most relevant, and higher feature values are related to higher model outputs. Intuitively, this is reasonable for this model, which estimates

recruitment, i.e. age-1 abundance. Interestingly, the observation data has greater importance than SAM estimates, indicating limited reliability of the SAM model, which is also reflected in the low $R^2$ values in Table 2.

For the SSB estimation model, the SAM-estimated SSB is most relevant, followed by the SD24_poundnet observations, which are not age-stratified and thus are also closely related to SSB. Here, in contrast to the recruitment model, the SAM estimate is the most important feature, indicating that the SAM model is more reliable, which is consistent with the high $R^2$ values of the SAM-based SSB estimates shown in Table 2.

In summary, the feature importances are intuitively reasonable and consistent with the performance evaluation, indicating that the ML models indeed capture the true, underlying stock dynamics.

## 4 RELATED WORK

ML methods are increasingly used to model ecosystem dynamics [14]. One motivation for using ML models is to use them as meta-models, to emulate computationally expensive physical models. For example, Rammer and Seidl [16] used a deep neural network to predict vegetation transitions, which has subsequently been used to forecast post-fire vegetation regeneration under different climate and fire regimes [15]. Another promising direction are hybrid models that use data-driven ML for modeling some system components, and mechanistical, knowledge-based models for others [18]. Such hybrid models have, for example, been used for estimating evaporation [8] or lake water temperature [17].

The main challenge in applying ML to ecological modeling lies in the need for large, labeled datasets, limiting the application of ML to domains where such datasets are available [14]. Modeling fish stock

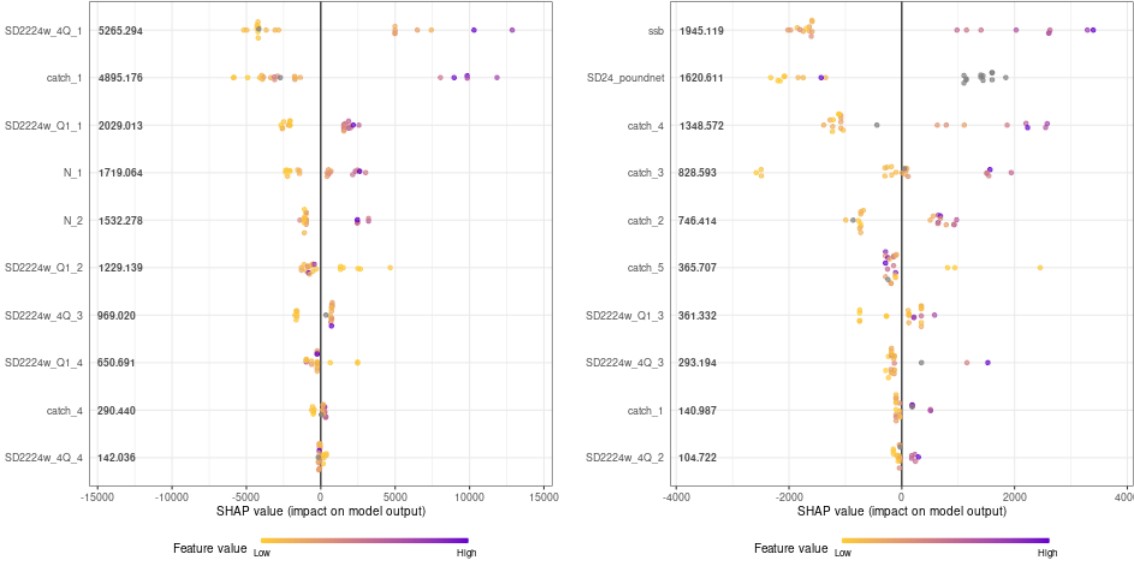

**Figure 3: SHAP feature importance values of recruitment estimation model (left) and SSB estimation model (right) for the Western Baltic cod stock. Here, SD2224w_4Q, SD2224w_Q1 and SD24_poundnet denote survey data, catch denotes commercial fisheries data, and N is the abundance estimated by SAM. The index denotes the age of the fish.**

is specifically challenging in that regard: A ground truth can usually not be obtained (e.g., the true stock size), and datasets are small (even for well-monitored stocks, yearly surveys lead to time series of < 50 years). Therefore, the adoption of ML for stock assessment has been limited. Previous studies [4, 9, 10, 19] focused on modeling the stock-recruitment relationship, a central but challenging aspect of stock assessment. These approaches use ML models (like naïve Bayes [4], small artificial neural networks [9, 19] or random forests [10]) to forecast recruitment, based on different environmental data (like sea surface temperature or salinity) and SSB. ML has also been applied to analyze the relationship between other stock parameters and environmental variables, like the effect of environmental variables on length-at-age of herring stocks [12].

All of these approaches use SSB estimated by a stock assessment model like SAM as a feature. However, stable SSB estimates are only available in retrospect, as the SSB estimate can be strongly influenced by future observations. Thus, it is not clear whether existing models can be used for forecasting stock parameters. Instead, the primary objective of these previous studies was to evaluate the predictive performance of environmental variables on stock parameters. In addition, these previous studies typically compare different hyperparameter configurations (e.g., the number of hidden nodes in a neural network) and report the model with the best test performance, leading to an overly optimistic performance assessment due to optimizing for the test data. In contrast, our work aims to forecast stock parameters using only data that is available at the time where the forecast is made, and we deliberately avoid hyperparameter optimization to obtain a more conservative performance estimate.

## 5 DISCUSSION AND CONCLUSION

In this paper, we proposed a hybrid approach for forecasting stock parameters. The approach uses an ML model to improve the accuracy of an initial forecast of a SAM model. Unlike previous studies that relied on retrospecive data, our model relies solely on data available in the assessment year, enabling its applicability for forecasting purposes. Our approach outperformed the baseline SAM model in a majority of the cases examined, particularly in forecasting SSB.

It is important to acknowledge several limitations of our study. First, the scarcity of data and the absence of ground truth stock parameters make interpretation of the results challenging. We intentionally refrained from conducting model comparisons or hyperparameter optimizations to provide a conservative estimate of our model's performance. Still, whether the promising performance of our approach will hold in practice, i.e., for other stocks or future assessment years, especially under qualitative changes of the stock dynamics due to climate change, is a question for future research. Furthermore, our current approach does not directly quantify the uncertainty of forecasts, which is crucial information for effective management decision-making. To address this limitation, it is necessary to explore probabilistic and calibrated ML methods that can propagate the uncertainty from the SAM estimates to the final predictions. Finally, our approach solely relies on survey and catch data, which are also the primary data sources for SAM models. Integrating additional features, e.g. environmental parameters such as sea surface temperature or salinity (which have previously been shown to influence stock dynamics [4, 9, 10, 19]) is an important avenue for future research.

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
