# OpenReview forum: "Towards Machine Learning-based Fish Stock Assessment"
_KDD.org/2023/Workshop/Fragile_Earth — KDD 2023 Workshop Fragile Earth Submission_

### Official Review · Reviewer_aMZv · 2023-07-10
**Review of "Towards Machine Learning-based Fish Stock Assessment"**

**Rating:** 8
**Confidence:** 3

**Review:**

This paper proposes a hybrid approach to improve the traditional fish stock estimation methods. The proposed method first uses the existing statistical methods (such as age-structured state-space models - SAM) to assess the fish stock, followed by correcting the SAM estimates using a gradient boosted tree based ML method. The authors provide experimental results from 5 different fish stocks and the results look promising from the paper.

Overall, the paper is easy to read, and the experimental results and the logic behind using an hybrid method is interesting. Overall, I support the acceptance of this paper for the venue.

A couple of feedback to improve the paper are below:
a. The experiments show the results from SAM and the proposed ML method. However, there are ML models (4, 9, 10) that may need to be compared against. I would be happy to see a comparison between pure statistical (SAM), pure ML and hybrid methods to make a sense of the impact. I anticipate with enough data, pure ML to be better but the Hybrid method should be a good bridge between the availability of training samples, and not.
b. Recruitment and SSB is not clear for me, I think some of these concepts may need a separate "key concepts" section. That will help the readers a lot.
c. Figure 3 is a very important one to understand the impact of different features. Instead of using these keywords, can you use better keywords for readers to immediately understand what these are? I know the captions reflect these but still it will be better to see them without explanation. Also, the figure deserves some explanation in the paragraph as well.
d. Section 4 (Related Work) and some parts of the introduction are similar. Why don't you move the related work right after introduction? This will help readers understand the novelty of the proposed method.
e. What are the different features that are being used by the related work with ML vs the proposed method? That will need to be emphasized. Also, what additional data would further improve? There should be some intuitions regarding these. Last sentence of the Section 5 starts explaining these, but maybe some intuition would help. For example "environmental parameters such as water temperature, etc."

---

### Official Review · Reviewer_CTxj · 2023-07-13
**Review for "Towards Machine Learning-based Fish Stock Assessment"**

**Rating:** 7
**Confidence:** 2

**Review:**

 Summary : A hybrid use of a machine learning model and more traditional statistical model is used to tackle the problem of Fish stock assessment. It has been shown that utilizing the power of ML to empower the traditional statistical models can provide for great improvement in estimates.

Strengths :
 - Application of ML method in solving a environmental challenge is sufficiently demonstrated.
 - The results have been shown and tested on multiple data sets .

Weaknesses :
 - Details and justification about use of Gradient Boosting is lacking.
 - Method has been tested against other ML methods for comparison.

Questions : On what basis were different feature subsets selected ? Were any feature selection/dimension reduction techniques such as PCA applied?

---

### Official Review · Reviewer_gedV · 2023-07-13
**Review for Towards Machine Learning-based Fish Stock Assessment**

**Rating:** 7
**Confidence:** 3

**Review:**

In this paper, the authors propose a hybrid model that uses gradient boosted trees as a post processing technique to improve the accuracy of the initial estimates from mechanistic models. The authors show the superior results of the proposed hybrid model with Retrospective estimates of recruitment and spawning stock biomass of five different stocks. The authors also show the usefulness of the hybrid model by presenting the SHAP feature importance values for the two task - recruitment and SSB.

---

### Official Review · Reviewer_zpmn · 2023-07-16
**Review for "Towards Machine Learning-based Fish Stock Assessment"**

**Rating:** 7
**Confidence:** 4

**Review:**

Summary:

In this paper, the authors investigate the use of machine learning models to improve the estimation and forecast of such stock parameters. More specifically,  the authors propose a hybrid model that combines classical statistical stock assessment models with supervised ML, specifically gradient boosted trees. Extensive experiments on five different stocks and find that the forecast accuracy of recruitment
and spawning stock biomass improves considerably in most cases.

Strengths :
- The introduction is covered thoroughly and differentiates the contributions of the paper well.
- Important and interesting problem.
- Experiments are comprehensive.

Weaknesses :
- Can the authors provide the standard deviation of the forecasting results in Table 1?
- More details of experimental setup should be provided.

---

### Decision · Program_Chairs · 2023-07-19

**Decision:**

Accept (Oral)

**Comment:**

Congratulations!

We are pleased to inform you that your submission: Towards Machine Learning-based Fish Stock Assessment has been accepted to The KDD 2023 Workshop Fragile Earth: AI for Climate Sustainability - from Wildfire Disaster Management to Public Health and Beyond.

Camera ready deadline is ** July 24 AOE **.  Please log in to OpenReview and prepare your camera-ready version based on the reviews. Formatting rules are the same as for the initial submission and submissions must adhere to KDD 2023 guidelines available at https://authors.acm.org/proceedings/production-information/taps-production-workflow.

Again, congratulations on the acceptance of your paper!  We look forward to seeing you at the workshop on Aug 7, 2023.

The Fragile Earth Workshop Proceeding Chairs